# Detection of Depression-Related Tweets in Mexico Using Crosslingual Schemes and Knowledge Distillation

**DOI:** 10.3390/healthcare11071057

**Published:** 2023-04-06

**Authors:** Jorge Pool-Cen, Hugo Carlos-Martínez, Gandhi Hernández-Chan, Oscar Sánchez-Siordia

**Affiliations:** 1Geospatial Information Sciences Research Center, Mexico City 14240, Mexico; 2IxM CONACyT, Mexico City 14240, Mexico; 3Laboratorio Nacional de Geointeligencia (GeoInt), Mexico City 14240, Mexico

**Keywords:** depression, text classification, knowledge distillation, dimensionality reduction, Twitter, COVID-19

## Abstract

Mental health problems are one of the various ills that afflict the world’s population. Early diagnosis and medical care are public health problems addressed from various perspectives. Among the mental illnesses that most afflict the population is depression; its early diagnosis is vitally important, as it can trigger more severe illnesses, such as suicidal ideation. Due to the lack of homogeneity in current diagnostic tools, the community has focused on using AI tools for opportune diagnosis. Unfortunately, there is a lack of data that allows the use of IA tools for the Spanish language. Our work has a cross-lingual scheme to address this issue, allowing us to identify Spanish and English texts. The experiments demonstrated the methodology’s effectiveness with an F1-score of 0.95. With this methodology, we propose a method to solve a classification problem for depression tweets (or short texts) by reusing English language databases with insufficient data to generate a classification model, such as in the Spanish language. We also validated the information obtained with public data to analyze the behavior of depression in Mexico during the COVID-19 pandemic. Our results show that the use of these methodologies can serve as support, not only in the diagnosis of depression, but also in the construction of different language databases that allow the creation of more efficient diagnostic tools.

## 1. Introduction

Mental health problems are an area of medical and social sciences that have become very important in recent decades, because the number of people who have suffered, or are suffering, a mental illness is increasing. Some studies estimate that almost one billion people worldwide have a mental disorder. Due to this, even on a global scale, multiple initiatives are trying to address mental health problems in a comprehensive way [1].

Due to the COVID-19 pandemic, many mental health problems have increased in recent years. Only a few years after the COVID-19 pandemic, it is possible to explore the effects of the pandemic on mental health. Recent studies suggest there has been a rise in mental health problems in people who were mentally healthy before the pandemic. On the other hand, people who had some previous condition prior to the pandemic have seen the effects of their mental illnesses increase [2,3]. In particular, the mental health of young people has drastically reduced [4].

Some mental illnesses have become so widespread among the population that they have become a subject of public health policy. In particular, depression is one of the leading causes of disability and can increase the risk of suicidal ideation and suicide attempts [5]. The latter has led to the creation of public policies that promote the treatment of depression in its early stages and the receipt of psychological and psychiatric care [6,7]. Like other diseases, mental health problems harm people’s well-being and directly impact activities of other natures, such as economic ones. For example, lost productivity due to two of the most common mental disorders, anxiety and depression, costs the global economy one trillion dollars annually [8].

In the case of Latin America, some studies suggest that 50% of people with depression do not receive adequate treatment, one of the leading causes being lack of diagnosis [9]. Some studies even suggest that a possible way to address the problem is by using the Internet to facilitate detection and treatment mechanisms [10]. Along the same lines, some studies suggest the use of apps to treat depression in Latino and Hispanic populations [11]. However, much work remains to be done. As we cite later, the literature promotes the creation of multilingual care schemes for Latino populations, with a particular emphasis on immigrants.

Depression is typically diagnosed based on individual self-reporting or specific questionnaires designed to detect characteristic patterns of feelings or social interactions [12]. However, these tools generally have some subjective components or are not applied homogeneously, which complicates the diagnosis process [13]. Due to the above, the opportune detection and diagnosis of mental illnesses have become very active research topics. The idea is to have more robust tools for early diagnosis that allow diseases to be treated promptly. From this idea, the use of computational tools for diagnosing and detecting mental illnesses has spread [14].

Machine Learning (ML), particularly Deep Learning (DL) algorithms, has successfully detected mental diseases and characterized behavior patterns. For depression, for example, there exist multidisciplinary solutions that use demographic and genetic information to improve antidepressant treatments [15], or applications based on Natural Language Processing (NLP) that successfully detect depression [16,17,18]. Since DL algorithms generally require a considerable volume of data, social networks have become an indispensable source of information [19,20]. In particular, Twitter has become a primary data source for feeding these algorithms [21,22]. However, one of the main problems is that the data sets are usually not public or homogenized, which often prevents reproduction of the results. In the case of NLP, the most used language processing models are those that are based on schemes such as Bidirectional Encoder Representations from Transformers (BERT) [23]. BERT-type models often lead to specific models for different languages. In Candida et al. [24] we find a general summary of the application of these models to mental health problems.

One of the under-researched areas in the detection and diagnosis of depression is the use of multilingual methodologies within the framework of NLP, which is the main idea of this work. We develop a methodology that allows the use of existing data sets of tweets in the English language to detect depressive tweets in Spanish. From a technical point of view, the detection of depression from Twitter posts requires the following two steps: the detection of tweets depression-related or that manifest depression; and the incorporation of a temporal component, which requires that users must publish tweets associated with depression with some frequency. The use of a temporal component is due to the fact that depression is a complex disease, the severity of which tends to vary over time. Therefore, it is necessary to consider the frequency of publications since it is impossible to determine the state of depression based on only a small group of publications. This is one of the weak points of using social networks to identify depression, as users must post texts with a certain regularity that allow the identification of a depressive state. In this work, we focused on the first element. Although our work did not focus on detecting depression, it is valuable as a first step in complete methodologies. We limited the scope of the current research because a complete methodology requires a database of user profiles diagnosed with depression by experts in the field and, unfortunately, this is a lack of such research in Spanish. In future investigations, we will address this issue.

The organization of the text is as follows. Section 2
presents related works that were taken as a reference for the present investigation. Section 3 introduces the theoretical framework used to develop the methodology. Section 4 details the methodology, based on the framework presented in Section 3. After that, in Section 5
and Section 6, we present the materials and experimentation schemes employed. In Section 7, we present the results obtained by our methodology and compare translations. Finally, Section 8 presents the results of applying our methodology to geo-referenced Twitter data in Mexico for the years 2018–2021.

## 2. Related Works

The use of NLP models in health problems has been a very popular topic of study. There are applications in the field of medicine in general [25,26]. In the literature, we find works that refer to the importance of creating multilingual schemes to address mental health problems. For example, in Brisset et al. [27], the authors describe the problems of providing primary mental health care to immigrants in Montreal due to language barriers. In Límon et al. [28], the authors highlight the problems in regard to early detection of depression in Spanish-speaking immigrants; in this research, the authors emphasized the problems of translating depression instruments from English to Spanish. In Garcia et al. [29], the authors mention that people with limited English proficiency are the ones who most frequently suffer from depression, mainly Latin American immigrants (see Figure 1).

In Figure 1, we can see the distribution of tweets related to depression in Spanish in the US border states, where much of the immigrant population is concentrated. The Figure illustrates the feasibility of using Twitter to monitor mental health issues in immigrant populations.

The detection of depression using social networks and learning algorithms is not new. Many works address the problem using different strategies. In particular, they can be distinguished by considering social networks that serve as sources of information, NLP models used to represent text, and classification algorithms used to distinguish factors. A complete description of existing works in the English language can be found in [30].

For the analysis of depressive tweets in the Spanish language, there are few works. Most focus on constructing dictionaries (or translated phrases) that include words related to depression, and then use these dictionaries to select depression-related tweets to generate statistical descriptors, or to train classification algorithms. For example, in [31], the authors introduce a comprehensive collection of Spanish words commonly used by depressive patients and gave insight into the relevance of these words in identifying posts on social media related to depression. One of the central affirmations of this work is that using dictionaries to identify post-depressive patients is inadequate, because the words are frequently used in different contexts. In Leis et al. [32], the authors present a methodology to identify signs of depression based on the linguistic characteristics of the tweets in Spanish. The authors selected Twitter users who indicated potential signs of depressive symptoms based on the 20 most common Spanish words expressed by patients with depression in clinical settings. Once users were selected, the authors used statistical descriptions of language and behavior to identify a sign of depression. In Valeriano et al. [33], the authors use a dictionary of English phrases translated into Spanish to identify tweets related to suicide. Once phrases were identified, a manual selection was made to differentiate tweets that could correspond to expressions of sarcasm, song lyrics, etc., and, then, a machine learning algorithm was trained to classify depressive tweets. In Shekerbekova et al. [34], the authors compare different machine learning algorithms to identify posts related to depression. As in our work, the authors selected a set of posts related to depression and general posts.

If the literature on identifying depression in the Spanish language is insufficient, it is almost null in the case of multilingual models. Moreover, most studies have a comparative approach, rather than considering it as a multi-language problem. For example, in Ramirez et al. [35], the authors use computational methods to compare expressions of depression in English and Spanish. It is a comparative study of variations in expressions of depression in both languages. There is research that, although formulated for the English language, implicitly used NLP models allow working with text in other languages. For example, in Basco et al. [12], the authors incorporate multilingual NLP models to detect depression and gambling disorders. The authors argue that many users generate posts in languages other than their native ones (e.g., English).

Some works intend to detect signs of depression regardless of the language used. For this, data from conversations in different languages and algorithms for extracting speech features are employed. For example, Kiss et al. in [36], evaluate the possibility of extracting speech characteristics as descriptors to identify depression. This work suggests that the descriptors found are similar regardless of language. On the other hand, in Demiroglu et al. [37], the authors use a combination of sound and text descriptors. For this, the extracted speech features are merged with sentiment analysis expressions obtained through text. Finally, in Kiss, G. [38], the author discusses and evaluates the possibility of generating models to identify depression using speech in different languages and assesses the ability to identify depression regardless of the language used.

To our knowledge, the works closest to ours are the ones presented in [39,40]. These papers present a methodology for detecting depression based on the construction of linear transformations that are capable of aligning words in different languages. For a set of equivalent words in both languages, it is possible to find a linear transformation *W* (viewed as an embedding space) that maps between languages. This transformation makes it possible to train a classification algorithm in a language (e.g., English) and use this classifier for texts in Spanish. Among the main differences from our work are the type of transformation used and the inclusion of attention mechanisms to maintain semantic properties. While in [39,40], the mapping is only between words, our methodology used knowledge distillation to find more complex mapping functions, while incorporating semantic properties.

It is important to note that using knowledge distillation to manage multilingual schemes is not the only viable option to identify or classify depression. There are models in the literature designed to handle multi-language sentences, such as that presented in Feng et al. [41]. Some works apply these models, for example, in sentiment analysis [42]. However, the results presented in Reimers et al. [43] showed a better vectorial representation of sentences in different languages. Due to this, in this work, we focused on applying knowledge distillation to detect tweets that were depression-related.

Derived from the literature review, we detected the following limitations. The models based on dictionaries or associated phrases restrict the detection capability to the quality of the dictionaries; furthermore, these schemes are not practical for multilanguage problems, since the dictionaries could vary significantly between different languages. On the other hand, the explicit use of translators only partially solves the problem, since the texts found on social networks are usually concise, and, in the translation, there may be a loss of context. Finally, the lack of data in other languages (besides English) complicates reproduction of the results. With these limitations in mind, the question arises as to whether it is possible to build a model trained with a limited amount of data and easily generalized to other languages without re-training or building new databases.

Our proposal arose as a response to these problems. The main idea was to build an embedded space containing phrases with similar semantic and syntactic content so that dictionaries or translations are not explicitly needed. This space could be used to train classification models in a specific language (e.g., English) to be used to detect similar phrases in other languages. One way to generate such a space is through knowledge distillation and dimensionality reduction schemes. The following section presents the necessary concepts to build this space.

## 3. Framework

As previously mentioned, we used knowledge distillation to obtain the vector representation of the tweets. Unfortunately, since they were usually concise texts, it was convenient to use a dimensionality reduction scheme; in particular, we used the proposal presented in [44] (known as IVIS). In the following sections, we describe, in a general manner, the mathematical foundations of both methodologies.

### 3.1. Knowledge Distillation

The general concept of knowledge distillation refers to the process of knowledge transfer from large models (i.e., a large number of parameters) to simpler models designed to perform specific tasks. These models are formulated in terms of teacher and student models. The idea is that the student model can be trained on specific tasks from the master model. These methodologies are trendy in NLP tasks, where large models have been trained with a large amount of data. In our work, we used the knowledge distillation methodology presented in [43]. This model proposes mapping a translated sentence of a language to the same vector space as the original language’s sentence to mimic the language’s properties, i.e., this knowledge distillation aims to extend one language’s characteristics or properties to another. In other words, the original and translated vector representations of semantically similar declarations must be neighbors.

As we mentioned earlier, the idea starts with a teacher model, denoted by *M* for a language *s*, and a parallel set of translated sentences, denoted $\left(\left(s_{1},t_{1}\right),\ldots \left(s_{n},t_{n}\right)\right)$; with $t_{i}$ being the translation of $s_{i}$. Then a model called Student, denoted by $\hat{M}$, is trained as $\hat{M}\left(s_{i}\right)\approx M\left(s_{i}\right)$ and $\hat{M}\left(t_{i}\right)\approx M\left(s_{i}\right)$, through the loss function:(1)$$\frac{1}{\left|\beta \right|}\sum_{j\in \beta }\left[{\left(M\left(s_{j}\right)-\hat{M}\left(s_{j}\right)\right)}^{2}+{\left(M\left(s_{j}\right)-\hat{M}\left(t_{j}\right)\right)}^{2}\right],$$
where β represents a batch of sentences. In the first part of the equation, ${\left(M\left(s_{j}\right)-\hat{M}\left(s_{j}\right)\right)}^{2}$, the student model learns to project the sentence onto the same vector space as the teacher model. The second part of the cost function, named ${\left(M\left(s_{j}\right)-\hat{M}\left(t_{j}\right)\right)}^{2}$, aims to teach the student model how to project the translated sentences to the exact location in the vector space as the original sentences. That is, sentences with similar semantic content are close (in Euclidean distance) regardless of language.

In practice, we used the model distiluse-base-multilingual-cased-v1 (DBM) formulated in [45] and implemented in [46]. This model is a sentence-transformers model; that is, it maps sentences and paragraphs to a 512-dimensional dense vector space and can be used for tasks like clustering or semantic search. The model was trained in 15 languages, including English and Spanish.

### 3.2. Dimensionality Reduction with Ivis

Text is an unstructured data type that can be encountered with various lengths, so extracting features in a corpus can generate high-dimensionality sparse vector representations. Using dimensionality reduction algorithms reduces the vector dimension while maintaining the quality of the vector representation of the original data. Several dimensionality reduction algorithms exist in the literature, such as PCA, LDA, t-SNE, IVIS, and ISOMAP [47]. However, IVIS has shown performance comparable to, or superior to, the algorithms mentioned above. IVIS was conceived as a Siamese neural network with a triple loss function. The results reported by the authors emphasize that IVIS preserves global data structures in a low-dimensional space for real and simulated data sets.

The IVIS algorithm is a non-linear dimensionality reduction method, based on a neural network model with three training schemes: supervised, unsupervised, and semi-supervised. The cost function used in training the neural network is a variant of the standard triple loss function.
(2)$$L_{tri}\left(\theta \right)=\left[\Sigma_{a,p,n}D_{a,p}-min\left(D_{a,n},D_{p,n}\right)+m\right],$$
where *a*, *p*, and *n* correspond to a sample of interest, a positive sample, and a negative sample, respectively. *D* is a distance function, and *m* is the margin parameter. The distance function *D* corresponds to the Euclidean distance and measures the similarity between the points *a* and *b* in the embedded space.
(3)$$D_{a,b}=\sqrt{\sum_{i=1}^{n}{\left(a_{i}-b_{i}\right)}^{2}}$$

The loss function minimizes the distance between the point of interest and the positive sample, while maximizing the distance to the negative sample. At each point of interest in the dataset, positive and negative samples are received according to the k-nearest neighbor algorithm.

We used IVIS for the conversion of vector representations with dimension 512 to two-dimensional representation vectors. This number of dimensions was selected because the results did not improve significantly in the experiments carried out when considering larger dimensions.

In the following sections, we present the proposed methodology. We first introduce the set of data used and then describe the characteristics of each stage.

## 4. Methodology

Identifying depression-related tweets was carried out in four stages: (1) pre-processing, (2) feature extraction, using knowledge distillation methodology, (3) dimensionality reduction, using IVIS and (4) tweet classification. In the following sections, we describe each of the stages and give a hint of the importance attached to its application.

### 4.1. Pre-Processing and Feature Extraction

In this phase, we processed each tweet to normalize the text to lowercase and removed the blank spaces found at the beginning and end. Next, we removed null records, duplicate records, emojis, hyperlinks, mentions, punctuation signs, and words that contained the symbol @ or #. During the cleaning process we removed records with single-word phrases that did not carry any meaning, for example, abbreviations such as thx and thd. It is essential to mention that after the pre-processing phase, the size of the datasets did not change significantly. Finally, we used the DMB model to obtain its vector representation.

### 4.2. Dimensionality Reduction

Extracting text features generates high-dimensional vector representations; however, these representations can be sparse vectors due to text features, such as the length of each text. The dimensionality reduction method helps to compress the information and maintain the qualities of the original data. As mentioned above, we used existing IVIS training schemes to evaluate our methodology, considering the problems encountered in practice. For example, a semi-supervised strategy can be used when one of the datasets contains a few unclassified tweets. On the contrary, an unsupervised strategy is usually used when there are no labels, but the text refers to fewer topics.

### 4.3. Tweet Identification

The ultimate goal of our methodology was to correctly identify tweets related to depression. In principle, after obtaining the 2D vector representation, it would be possible to apply a simple classification algorithm. In particular, we compared the results obtained by the following algorithms: Logistic regression (LR), Support Vector Machines (SVM), Gaussian Process (GP), and Quadratic Discriminant Analysis (QDA). The idea was to evaluate whether our methodology was robust, regardless of the classification algorithm. For all classifiers, the hyperparameters were determined using a grid search cross-validation strategy, and experiments were performed using the Scikit Learn library [48].

## 5. Materials and Methods

In this work, we considered three possible classes. The first class, $C_{D}$, labeled as 1, corresponds to tweets related to depression. The second class, $C_{N}$, labeled as 0, corresponds to tweets that are not related to depression. Finally, the third class, $C_{U}$, corresponds to tweets with unknown content. We used the $C_{U}$ class to evaluate semi-supervised dimensionality reduction methods.

We created four data sets, all of which contained phrases related to the topic of depression: $D_{1}$, $D_{2}$, $D_{3}$, and $D_{4}$. Some texts might contain news or reports on depression, while others were posted by users who expressed depressive emotional feelings. The data set $D_{1}$ was obtained from Kaggle (https://www.kaggle.com/general/234873, accessed on 22 December 2022) and consisted of 4493 tweets in English, of which 2385 were tagged with class $C_{D}$ and 2263 with class $C_{N}$. The data set $D_{2}$ contained 2000 Spanish tweets published in 2019 extracted from the AGEI platform (http://agei.geoint.mx/, accessed on 9 November 2022), with 50% of the data corresponding to tweets related to depression; all tweets were labeled by experts. The data set $D_{3}$ contained 5093 tweets and was made up of a mixture of $D_{1}$ and 600 tweets randomly obtained from $D_{2}$. The data set $D_{4}$ was a subset of $D_{2}$ and contained 1400 tweets distributed in 50% for the depression class and the other 50% for the non-depression class. This data set was used as a test for the semi-supervised dimensionality reduction experiment, explained in Section 4.2. Table 1 shows the results of the exploratory analysis of the texts with respect to the length and number of words.

## 6. Experiments

We divided the experiments according to the dimensionality reduction methodology employed. Specifically, we designed the experiments according to supervised, semi-supervised, and unsupervised methodologies. This was because each methodology represents a different approach to the problem encountered in practice when finding depression-related tweets.

### 6.1. Supervised Dimensionality Reduction

Once the vector representations of the data sets $D_{1}$ and $D_{2}$ were obtained, we trained IVIS using the supervised scheme. In this series of experiments, we used only the data set $D_{1}$ for the training phase, that is, IVIS and the classification algorithm were trained only on English data. The idea was to evaluate whether it was possible to use depression-related tweets written in English to detect tweets with similar content in Spanish.

### 6.2. Unsupervised Dimensionality Reduction

We trained IVIS and the classification algorithms for these tests using the data set $D_{1}$ without including labels. The idea was to assess whether the methodology was robust when there were no labeled data, but one of the topics (in this case depression) was predominant. This experiment could be understood if we assumed that the syntax of depression-related tweets has a semantic structure that makes it possible to differentiate them from other topics (i.e., not depressive).

### 6.3. Semi-Supervised Dimensionality Reduction

This experiment’s training and test data corresponded to the data sets $D_{3}$ and $D_{4}$, respectively. For these experiments, we evaluated the ability of our methodology to assign labels to data that could be mislabeled. On many occasions, when evaluating whether a tweet is depression-related, there may be discrepancies between experts when labeling it. One way to address this problem is to leave these tweets unlabeled, letting the methodology assign the corresponding class from its vector representation. On the other hand, in some cases, if the dataset of the language of interest contains little data, it might be convenient to use the semi-supervised methodology.

The experimentation phase was carried out in the months of October and November of 2022 on a computer with i5 at 4.10 GHz and 16 GB RAM on OS Debian.

### 6.4. Experiments with Translations

Although it is a naive idea, the use of translations to identify tweets in different languages has been frequently used in other problems, such as sentiment analysis. We compared this strategy using translations obtained through the Google Translate platform using English phrases from the data set $D_{1}$ as a source of information. Once we obtained the translations, we used the BETO model to obtain the vector representation [49]. In other words, in this set of experiments, the only language used was Spanish. To do this, we built data sets in Spanish from the original sets $D_{1}$ and $D_{3}$ and used the BETO model to build the vector representation of all tweets (including translations). Throughout the document, we distinguish between tweets written in Spanish (i.e., native Spanish) and the translations obtained with Google. Table 2 summarizes the contents for each data set used, with translations and knowledge distillation.

The experimentation phase was carried out in October and November of 2022 on a computer with an i5 processor at 4.10 GHz and 16 GB RAM on OS Debian. This work was part of the Self-inflicted Death Study Seminar (SIEMAI) (http://siemai.geoint.mx/, accessed on 22 December 2022), with the voluntary collaboration of mental health experts.

## 7. Results

In this section, we present the results obtained from the experiments. To compare the performance of the classification algorithms, we used the following metrics: accuracy, precision, recall, and F1 metrics.

### 7.1. Evaluation of Experiments with Translations

This section presents the performance measures of the classification models using translations, the extraction of text features using the BETO model, and the various dimensionality reduction schemes. Table 3 shows the results obtained for this strategy. In these experiments, the best score was obtained with unsupervised dimensionality. The best models were Logistic Regression and Linear SVM with 0.85 on the F1-Score. In general, there did not seem to be any significant difference between the different dimensionality reduction schemes, which was understandable if we consider that, during translation, there were changes in the syntax that made classification difficult.

Figure 2 shows the classification results obtained using QDA. Note that the unsupervised scheme presented considerable dispersion, although, in general, it had the highest classification percentages. On the contrary, in the supervised and semi-supervised schemes, the data were in more compact regions but overlapped, which explains the classification percentages obtained. In the same sense, we must emphasize that adding a priori information about the classes did not seem to provide any significant advantage when using translations.

### 7.2. Evaluation of Experiments with Knowledge Distillation

Table 4 shows the results of classifying depression-related tweets using knowledge distillation. The best results for the supervised and unsupervised schemes were obtained by GP, with an accuracy and an F1 score of 0.93. Depression-related tweets could be classified using this model with reasonable accuracy.

Finally, the semi-supervised scheme obtained very high accuracy percentages for all the classifiers, which could be explained if we consider that few tweets in Spanish were included during the training. Concerning the F1 score, the best results were obtained by Logistic Regression; however, the differences between the classifiers were insignificant.

In Figure 3 we show the results obtained using QDA. In the figures, the boundary surfaces were constructed using the training data (i.e., tweets in English). Note that the data was much more concentrated for the semi-supervised scheme, while the supervised and unsupervised schemes were much more dispersed.

## 8. Geospatial Analysis of Depressive Tweets in Mexico

We employed our methodology to perform a space–time analysis of the depressión-related Tweets obtained through the AGEI platform. For this analysis, we only used public tweets containing geo-referenced information because we wanted to identify the State and date of publication. The objective was to analyze the tweets’ content, State of publication, and dates to compare the information with official data published in the same period. We built two descriptors based on geo-reference and user IDs. The first descriptor corresponded to the rate of tweets per State, and we built it using our methodology with the semi-supervised IVIS scheme and QDF as a classifier. Once we identified tweets with depression-related content, we used the user ID to generate a rate of user accounts that posted these tweets. We defined the State to which each user belonged depending on the State wherefrom the user posted most frequently; this was because some users posted in different States over time. Both rates described the rate per 100,000 inhabitants and were estimated using the information corresponding to the INEGI Population and Housing Census for 2020.

### 8.1. Analysis during the COVID-19 Pandemic Period

As we previously mentioned, the COVID-19 pandemic caused changes in the population’s behavior patterns. Although there are studies on the effects of the COVID pandemic in Mexico, we used our methodology to capture the variations in the publication of depression-related Tweets on a time scale that we divided into two periods. The first period corresponded to 2018–2019, which practically enclosed the interval before the pandemic. The second period corresponded to 2020–2021, when the pandemic had its most significant peak.

#### 8.1.1. Tweet Distributions

One of the most significant aspects to study during the pandemic was the change in behavior due to long periods of confinement. As a first analysis, we used tweets with content related to depression to analyze behavioral changes, especially in the periods of the highest contagion. To do this, we identified the date and place of publication and calculated the monthly distribution for each period.

Figure 4 shows the distribution of tweets related to depression for the different periods. The distributions illustrate the change in behavior in the publication of tweets. In the 2018–2019 period, the publications seemed to be more evenly distributed throughout the year, while, for the 2020–2021 period, the distribution shifted to the left, which corresponded to the second quarter of the year, months in which the highest COVID infection rates occurred.

#### 8.1.2. Content Analysis

To describe the content of the tweets related to depression in the evaluated period, we used the importance scores of each word obtained through TD–IDF. The main idea was to identify which words were commonly used each year.

The results can be seen in Figure 5. The results showed a change in the most relevant words in the years evaluated. For example, in 2018, the most relevant words referred to concepts related to the family, parents, etc. On the other hand, 2019 showed that words related to security and violence were gaining more importance. For the 2020–2021 period, the terms associated with the pandemic became essential. Note that the words that referred to family and parents remained relevant for all years.

### 8.2. State of Mood of Twitter Users in Mexico

Among the indicators published by INEGI is one associated with the state of mind of Twitter users in Mexico. INEGI calls this indicator the positivity rate and defines it as the number of positive tweets divided by the number of negative tweets for a given geographical area for a given period. Using depression-related tweets, we calculated an equivalent ratio by dividing the number of non-depression-related tweets by the number of depression-related tweets. We illustrated the behavior of both curves for the States with the highest suicide rate in Mexico.

The results can be seen in Figure 6. Note that, for the four States, both curves maintained the same trend. In the cases of Mexico City and Aguascalientes, the curves had very similar measurements. On the contrary, in Yucatán, although the trend was the same, the ratio between Tweets was generally above the curve. On the other hand, Coahuila was the opposite of Yucatan; in this State, the positivity rate was generally above the depression rate.

### 8.3. Depression in MéXico

To assess the ability to use the information from Twitter as an indicator of depression levels in Mexico, we compared the rates of Tweets and User Accounts against the official data provided by INEGI. Unfortunately, there were few official data on depression in Mexico; the existing data corresponds to depression rates per 100,000 inhabitants published by INEGI in 2021. However, there are official rates related to suicide, which we included in our analysis to make it more complete. First, we estimated the correlation between these official data against the user’s accounts and tweet rates for the different States.

The correlation analysis can be seen in Table 5. The results show a weak negative correlation between suicide and depression rates published by INEGI; this is relevant because, in many studies, this behavior suggests an under-reporting of depression cases at the national level. However, there was a negative correlation between depression rates and Tweets and User Accounts. A significant positive correlation could also be observed between the descriptors used and the published suicide rates. The results show that it is possible to use the rates of Tweets and User Accounts as an auxiliary estimator in the construction of national measures of suicide.

## 9. Discussion

During this study, we highlighted the importance of the fact that depression, as a mental health problem, can lead to other more serious problems, such as suicide, which is considered one of the leading causes of death in young people around the world. With this in mind, and from the existing limitations of the models and methodologies reported in state-of-the-art, our study proposes a method to generate classification models using the knowledge distillation technique.

Our results showed that the explicit use of translations in short texts reduces the accuracy of text classification, because there is a loss of context when translating short texts. On the other hand, the results using knowledge distillation showed better performance than translations, even with unlabeled data. The dimensionality reduction schemes used in this research generated similar vector representations, regardless of the use of labels. This allows models to be trained even with unlabeled or few data. Our results also illustrated the robustness of the methodology; regardless of the classification method used, the values in the f1-score were comparable.

We applied the model with the highest f1-scores to analyze tweets published in the period 2018–2021. We selected these years in order to determine the effect of the COVID-19 pandemic on the mood of the population in Mexico. The findings showed that the pandemic affected the mood of the Mexican population and that this was reflected in an increase in depression levels, which coincided with the results of the National Self-Reported Well-being Survey (ENBIARE). This scenario allows us to assume that, based on geospatial analysis, it is possible to have an approximation of the state of mental health in Mexico, state by state, to detect risk zones, and even to detect other factors that could be associated, almost in real-time.

## 10. Conclusions

The existing depression-related tweet classification models in the literature have three fundamental disadvantages: the use of dictionaries, possible loss of context, and the lack of generalization to multilanguage schemes. This work presents a methodology that responds to the existing state-of-the-art limitations. In particular, this work presented a methodology to classify depression-related tweets in Spanish. The methodology uses knowledge distillation and dimensionality reduction to train classification algorithms that allow the distinguishing of tweets related to depression in English and Spanish. The results obtained by the unsupervised schemes show that it is feasible to apply our proposal even in the absence of labeled data.

One of the drawbacks of this work is that it requires two stages. The first is in control of building the embedded space where the sentences in different languages are represented, and the second one uses dimensionality reduction algorithms. This has a computational cost that could be reduced; for example, designing a model that performs both tasks simultaneously. Furthermore, we tested this methodology only for short texts. Applications with long texts from other sources, like forums or news, require different models.

With this classification model, we aimed to generate a database of depression-related tweets to detect depression in its early stages and other related mental illnesses. Future work will apply this same methodology to other problems, such as suicide, misogyny, or bullying topics in Twitter. Since it is difficult to determine the state of mental health of a person with only some of their publications, as future work, we intend to expand this study to analyze the timeline of some Twitter accounts in order to generate models that allow us to identify individual cases of risk, but also, to be able to take advantage of the geospatial component (location of the person), to define prevention strategies.

## Figures and Tables

**Figure 1 healthcare-11-01057-f001:**
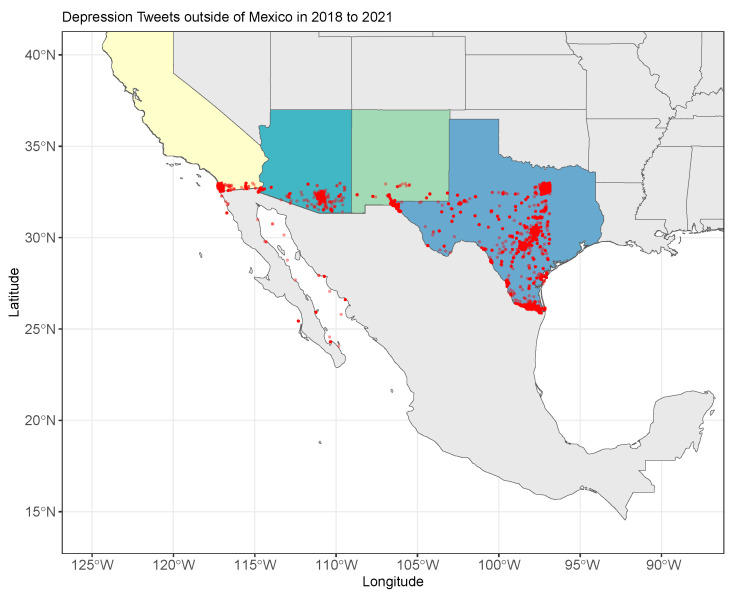
Tweets depression-related in Spanish in the United States border States from 2018–2021. Latin American immigrants are a population that frequently suffers from depression.

**Figure 2 healthcare-11-01057-f002:**
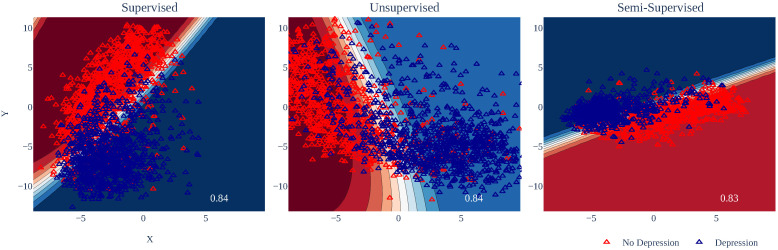
Results using translated sentences. Note that there was a substantial overlap between the different classes, which made the classification process difficult.

**Figure 3 healthcare-11-01057-f003:**
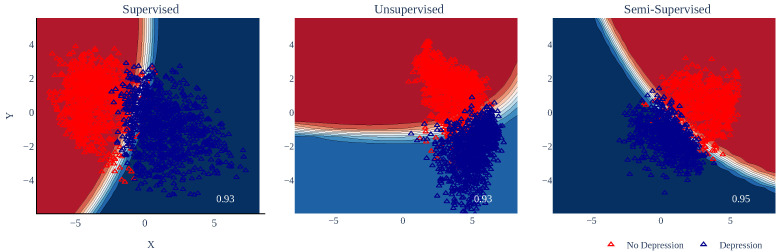
Results using knowledge distillation. Note that, regardless of the dimensionality reduction scheme employed, there was little overlap between the different classes.

**Figure 4 healthcare-11-01057-f004:**
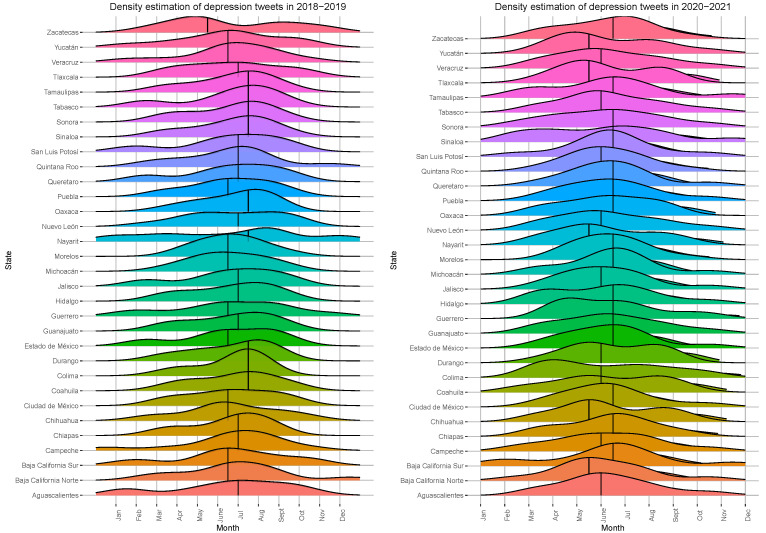
The density of tweets related to depression per month for the periods 2018–2019 and 2020–2021. It is possible to observe an increase in tweets for the second quarter of 2020–2021, consistent with the dates of the highest COVID-19 contagion in Mexico.

**Figure 5 healthcare-11-01057-f005:**
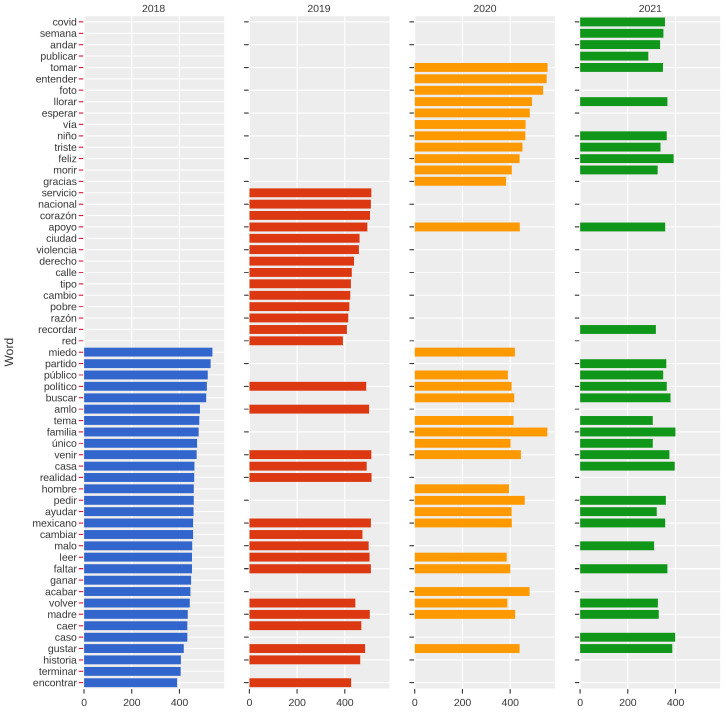
Importance of words for the period 2018–2021 using TI–IDF. For each year, the tweets related to depression refer to different topics. Note that, for the period 2020–2021, words associated with the pandemic became more relevant.

**Figure 6 healthcare-11-01057-f006:**
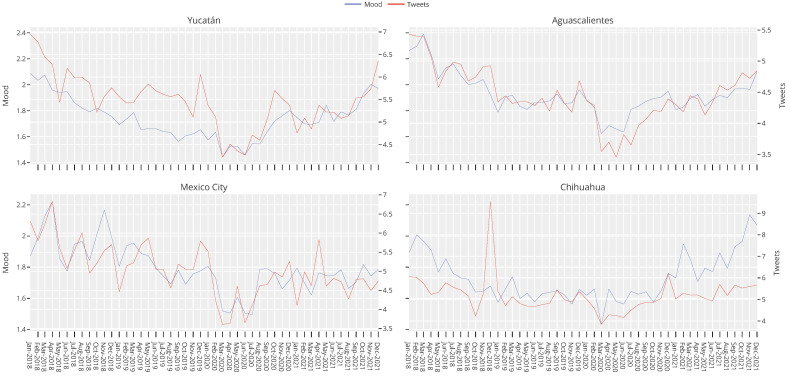
Comparison between the positivity rate, estimated by INEGI, and the depression ratio, estimated using the information obtained from Twitter. Note that the curves generally had the same trend.

**Table 1 healthcare-11-01057-t001:** Statistics for characters and words for datasets $D_{1}$–$D_{4}$.

Statistics	$D_{1}$	$D_{2}$	$D_{3}$	$D_{4}$
Characters				
Maximum	874	278	460	277
Minimum	9	7	7	8
Average	117.5	98.64	97.79	99.66
Words				
Maximum	73	57	73	57
Minimum	2	2	2	1
Average	17.22	18.38	17.3	18.64

**Table 2 healthcare-11-01057-t002:** Datasets used for the training and testing of the dimensionality reduction and classification models.

Datasets	Language	Samples	$C_{D}$	$C_{N}$	$C_{U}$
Knowledge distillation					
$D_{1}$	English	4648	2385	2263	0
$D_{2}$	Spanish	2000	1000	1000	0
$D_{3}$	English & Spanish	5093	2685	2263	600
$D_{4}$	Spanish	1400	700	700	0
Translations					
$T_{1}$	Translations	4648	2385	2263	0
$T_{2}$	Spanish	2000	1000	1000	0
$T_{3}$	Translations & Spanish	5248	2685	2563	600
$T_{4}$	Spanish	1400	700	700	0

**Table 3 healthcare-11-01057-t003:** Results using translations.

IVIS	Model	Accuracy	Precision	Recall	F1
Supervised	LR	0.82	0.84	0.81	0.82
	SVM	0.82	0.84	0.81	0.82
	GP	0.82	0.81	0.81	0.82
	QDA	0.84	0.84	0.83	**0.83**
Unsupervised	LR	0.84	0.84	0.85	**0.85**
	SVM	0.84	0.84	0.85	0.85
	GP	0.84	0.84	0.85	0.84
	QDA	0.84	0.83	0.84	0.84
Semi-supervised	LR	0.82	0.82	0.84	0.84
	SVM	0.83	0.83	0.83	0.83
	GP	0.82	0.82	0.83	0.83
	QDA	0.83	0.82	0.86	**0.84**

**Table 4 healthcare-11-01057-t004:** Results for Knowledge Distillation.

IVIS	Model	Accuracy	Precision	Recall	F1
Supervised	LR	0.91	0.98	0.84	0.90
	SVM	0.90	0.99	0.80	0.89
	GP	0.90	0.98	0.82	0.89
	QDA	0.93	0.97	0.89	**0.93**
Unsupervised	LR	0.92	0.93	0.90	0.91
	SVM	0.92	0.93	0.90	0.92
	GP	0.91	0.94	0.88	0.91
	QDA	0.93	0.92	0.94	**0.93**
Semi-supervised	LR	0.95	0.96	0.94	**0.95**
	SVM	0.95	0.96	0.93	0.95
	GP	0.95	0.96	0.93	0.94
	QDA	0.95	0.96	0.93	0.94

**Table 5 healthcare-11-01057-t005:** Correlations between the rates of depression published by the INEGI and the descriptors obtained with Twitter.

Rate		Correlation	*p*-Value
Suicide	Depression	−0.3448455	$7.23\times 10^{-2}$
Suicide	Tweets	0.5186857	$4.68\times 10^{-3}$
Suicide	User accounts	0.4935746	$7.60\times 10^{-3}$
Depression	Tweets	−0.4858599	$8.76\times 10^{-3}$
Depression	User accounts	−0.4750278	$1.06\times 10^{-2}$

## Data Availability

The source files and datasets used during this research are available in: https://github.com/jpoolcen/classification-tweets-depresive, accessed on 22 December 2022. The repository includes the datasets and the codes for data processing. Even when we use data from twitter accounts, they are public data and the personal data was not used, so it is not part of this study.

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
