# Peer review of "Detection of Depression-Related Tweets in Mexico Using Crosslingual Schemes and Knowledge Distillation"

_healthcare, 2023, doi:10.3390/healthcare11071057_

Round 1

Reviewer 1 Report

Dear Authors,
Thank you for submitting your paper to the "Healthcare" magazine. I read it with great interest. It has many strong points. I would include there: a very socially relevant and moving topic, clear structures and a solid research section. I believe that your paper has significant publication potential. It still needs to be refined so that it can be directed further.

Please, see a list of my concerns below.

Introduction:
- the placement of the graphic (figure 1) is too early; it precedes the content yet to come. I suggest either deleting it or moving it to a more appropriate place.
- I would ask you to briefly explain why you think such posts appear so frequently on Twitter.
- Please explain why why you do not include the second aspect while "detecting depression from Twitter posts requires two elements"

- at the end, it would be good to include information on how the text is divided up, what it consists of and what each chapter is about

Related work
- this is an exciting part of your text. However, it would be helpful, based on the literature review, to derive research questions that are not currently addressed in the text (these should be based on the research gaps observed and should also be answered in the conclusions)

Materials and Methods // Methodology // Experiments - I would suggest, for readability and to maintain cause-and-effect relationships, to swap these sections. The methodological description has a broader meaning; it is from its assumptions about what and how you will research later.

In other words, I would suggest first describing the methodological approach (methods, tools, techniques) and only then the materials obtained. The order could look like this: the methodology with the experimenters, identifying depression-related tweets was carried out in four stages, and then the materials and methods.

In addition, it would be helpful to add the implementation date and the team that carried out the research.

It would be worth embedding the discussion in the secondary literature, explicitly pointing out the strands that your study develops. The conclusion, in turn, should be considerably expanded with an apparent reference to the aim and research questions of the material. I would also point out the limitations of the research and more strongly accept the contribution to the theory and the possible practical application of your research concept.

And in the end - please, check the language and edition of the paper (no spaces, double spaces, punctuation problems). 

Reviewer 2 Report

The figures should appear after citations in the text—for example, Figure 1 appears before Page 3 of the PDF file.

I recommend the authors use the following citation format when discussing specific studies. For example, Brisset et al. [25] describe the problems ...

Please explain Figure 1 after the citation in the text.

The results regarding comparing algorithms' performance could be discussed in more detail (Section 9).

Spanish; Our --> Spanish. Our

Reviewer 3 Report

Excellent work, I commend the authors on this contribution. The study delivers interesting insights in general. The methodology is largely clear, and the manuscript is well-written as well. However, I would like to offer some suggestions for improvement in the next version, please.

(1)

In order to provide a better understanding of the current state of NLP in healthcare, I recommend positioning the introduction appropriately within the context of recent studies that have implemented BERT-based models to extract embeddings or knowledge from free-text data. For example:

https://doi.org/10.5220/0011012800003123

(2)

If the Hugging Face repository was used, please ensure to cite their reference.

Wolf, T., Debut, L., Sanh, V., Chaumond, J., Delangue, C., Moi, A., ... & Rush, A. M. (2019). Huggingface's transformers: State-of-the-art natural language processing. arXiv preprint arXiv:1910.03771.

(3)

The original reference of Scikit-Learn library should be as below:

Pedregosa, F., Varoquaux, G., Gramfort, A., Michel, V., Thirion, B., Grisel, O., ... & Duchesnay, E. (2011). Scikit-learn: Machine learning in Python. Journal of Machine Learning Research, 12, 2825-2830.

(4)

Please provide further elaboration on the possible limitations of the study results.

(5)

Lastly, the conclusion could benefit from a more conclusive summary of the key takeaways from the article, rather than reiterating the methodology.
